# E(3) EQUIVARIANT SCALAR INTERACTION NETWORK

## ABSTRACT

Equivariant Graph Neural Networks have demonstrated exceptional performance in modeling geometric data frequently observed in natural science research. The fundamental component of such models is the equivariant operation, which involves operations such as tensor product and scalarization. We present a conceptual framework that unifies the equivariant operations via equivariant basis decomposition. Within this framework, we generalize the idea of replacing the equivariant basis with input features to design efficient equivariant operations capable of modeling different type-$l$ features. To implement this, we propose Scalar Interaction and design an equivariant network, Scalar Interaction Network (SINet), with it. SINet's efficacy extends to efficiently mapping high type-$l$ features while maintaining a complexity $O(L^2)$ with the maximum $L$, representing a significant improvement over the $O(L^6)$ of tensor-product methods. Empirical results demonstrate SINet's capability to model complex quantum systems with high precision and computational efficiency. Its performance is competitive with current state-of-the-art methods in the field, showcasing its potential to advance the modeling of geometric data. This work highlights the potential of scalar interaction as an building block for constructing equivariant networks and opens up new avenues for future exploration in these vital fields.

## 1 INTRODUCTION

Machine learning has emerged as a pivotal tool in natural science research in recent times, driven by its expansive application in quantum chemistry (Schütt et al., 2019), computational physics (Kochkov et al., 2021), and structural biology (Jumper et al., 2021). These disciplines have stimulated the exploration of geometric data (Joshi et al., 2023), primarily due to the Euclidean spatial distribution and intrinsic symmetry often inherent in their task-related data (Rupp et al., 2012; Berman et al., 2000). Consider an instance where one is predicting the force exerted by each atom within a specific atomic system configuration (Chmiela et al., 2017): any translation or rotation of the reference coordinates will proportionally transform the force exerted by each atom. To learn from such symmetric data efficiently necessitates the utilization of models fortified with built-in equivariance assurances.

In mathematical terms, such correspondence is articulated as group equivariant mapping (Esteves, 2020). For geometric graph modeling, neural networks constructed in accordance with this equivariance have been structured around tensor-product (Weiler et al., 2018; Anderson et al., 2019; Thomas et al., 2018; Fuchs et al., 2020). The intermediary features of these models consist entirely of direct sums of the representation space of irreducible representations. These methods employ a flexible framework to approximate equivariant operations with higher-order irreducible representations by executing tensor product between the feature and a learnable steerable kernel, coupled with Clebsch-Gordan decomposition (Weiler et al., 2018). Nevertheless, the computational complexity of such operations is elevated due to the costly tensor-product operation. Regrettably, the complexity escalates swiftly in relation to the order of irreducible representations. A different trajectory of research seeks efficiency via scalarization (Satorras et al., 2021; Villar et al., 2021). While these methodologies are significantly quicker compared to tensor-product, they can only manage scalars and order 1 irreducible representation (vectors in $\mathbb{R}^3$) features, rendering higher-order geometric tensors nonviable.

In this study, we articulate a conceptual framework for constructing equivariant operations through the perspective of equivariant basis decomposition. In essence, our approach reframes the construc-

tion of extant equivariant operations as the selection of an appropriate basis and weight function. Our framework successfully amalgamates the majority of existing methods under two broad categories, fostering a unified understanding of these operations. We spotted that the application of higher-orders in the feature is not necessary in contradiction to the efficiency. Instead, we propose using input fragment as the equivariant basis. Thus, we develop a novel equivariant operation which model interactions between different irreducible representations via scalar interaction. Importantly, our Scalar Interaction Network (SINet) adeptly manages higher-order representations with its complexity increasing in a quadratic manner, signifying a marked advancement over previous methods.

Finally, we evaluate SINet on QM9 (Rupp et al., 2012) and $N$-body system (Kwon et al., 2010), demonstrating its remarkable performance and efficacy. Our comprehensive analysis provides clear evidence of SINet's capability to model complex quantum systems with high precision, achieving state-of-the-art comparable performance in terms of both accuracy and computational efficiency. The robustness of SINet also extends to larger $n$-body systems, illustrating its scalability and versatility in handling quantum systems of varying complexities. These results collectively point to SINet as a valuable tool for advancing computational quantum chemistry and physics, bridging the gap between theory and computational capability, and paving the way for future investigations in these critical fields.

## 2 SE3 EQUIVARIANT OPERATIONS

Deep neural networks (DNNs) are composed of elementary operations. For example, Multi-Layer Perceptrons (MLPs) are combinations of linear transformations and element-wise non-linearity, while Convolutional Neural Networks (CNNs) are composed of linear convolution and non-linearity. As for the equivariant neural network, these operations need to be specialized for preserving the symmetry of features and eventually producing the output with a given requirement on the equivariance. In this section, we will describe the constraint on these operations and introduce two main operations for describing the interaction between two steerable features.

### 2.1 PRELIMINARY GROUP REPRESENTATION

Before delving into the details of steerable operations, we first provide a brief introduction to the steerable features introduced by (Thomas et al., 2018; Weiler et al., 2018; Worrall et al., 2017), which are rooted in group representation theory. Consider a regression task with input $x \in X$ and label $y \in Y$. As the coordinate system is rotated or translated, the input transforms as $x \to D^X(x)$ and the target transforms as $y \to D^Y(y)$. This change of coordinate system can be represented by an element $g$ from a group $G$, where $G$ is the $SE(3)$ group for describing rotation and translation in 3D space. The translation of $x$ and $y$ is described by a group *action* $g$ of $G$ on $X$ and $Y$. Moreover, if the mappings $D^X$ and $D^Y$ are linear transformations, they are called *representations* of group $G$. Two representations, $D_1$ and $D_2$, are considered *equivalent* if there exists a matrix $T$ that satisfies Equation 1 for all $g \in G$.

$$D_1(g) = TD_2(g)T^{-1}. \tag{1}$$

According to the Peter-Weyl Theorem (Peter & Weyl, 1927), any finite-dimensional representation is equivalent to a direct sum of *irreducible representations*. For the 3D rotation group $SO(3)$, these irreducible representations are known as *Wigner-D matrices*, denoted by $D^l(g)$. $D^l(g)$ is a $(2l+1) \times (2l+1)$ unitary matrix that acts on a $(2l+1)$-dimensional space, which is referred to as a *type-l vector* space $V^l$. Equivariant neural networks deal with *steerable features* in these spaces. Type-0 vectors correspond to scalars, while type-1 vectors represent normal vectors in 3D space, such as velocity and acceleration. Higher-order type-$l$ vectors may also arise in specific applications such as predicting higher-order electronic dipole moment. Since the direct sum of different Wigner-D matrices, $D(g) = D^{l_1}(g) \oplus D^{l_2}(g) \oplus \cdots$, is still a (reducible) representation of the $SO(3)$ group, the direct sum of several type-$l$ vectors, $h = h^{l_1} \oplus h^{l_2} \oplus \cdots \in H$, can also be used as steerable features. We denote the vector space of steerable feature $h$ by

$$H = n_0 V^0 \oplus n_1 V^1 \cdots \oplus n_L V^L, \tag{2}$$

where $L$ is the maximum degree of $l$ and $nV := \bigoplus_i^n V$ is the direct sum of $n$ copies of the same space $V$. In equivariant neural networks, a common practice is to assign the same number $n_i = c$

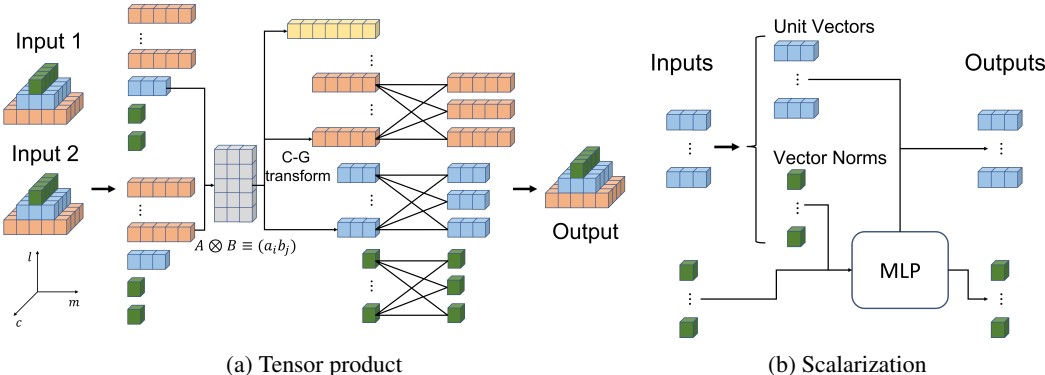

(a) Tensor product                               (b) Scalarization

Figure 1: The illustration presents the tensor product and scalarization operations. In (a), the tensor product is demonstrated between $h_1$ and $h_2$, which belong to $3V^0 \oplus 3V^1 \oplus 3V^2$. Specifically, the interaction between a type-1 vector and a type-2 vector is shown, where they are combined using the tensor product and subsequently decomposed into different vectors through CG-transform. This process is repeated for each pair of such vectors, followed by a linear combination. (b) depicts the scalarization process, which exclusively takes type-$l$ vectors and scalars as input. In scalarization, vector norms are fed into a multi-layer perceptron (MLP) alongside the scalars. The MLP then produces scalars that can be further combined with the input vectors to construct the output.

for steerable features in the hidden layers. This particular value of $c$ is commonly referred to as the *channels* of the steerable features.

## 2.2 EQUIVARIANT OPERATION

In this section, we give a formal definition for equivariant operations used in equivariant neural networks. Mathematically, a function $\mathcal{L} : X \rightarrow Y$ is *equivariant* with respect to a group $G$ if $\mathcal{L}$ commutes with group action for all $g \in G$, as shown in Equation 3.

$$\mathcal{L} \circ D^X(g)(x) = D^Y(g) \circ \mathcal{L}(x). \tag{3}$$

An additional conventional requirement for equivariant operations is that both the input and output features are steerable. This implies that the group representation on the input and output spaces should be a direct sum of Wigner-D matrices. Consequently, some mathematically equivariant functions, such as reversible linear transformations $x \rightarrow Ax$, become impractical for use in equivariant neural networks as it change the representation on input space $D^X(g)$ to an equivalent representation $AD^X(g)A^{-1}$, which in general is not the direct sum of Wigner-D matrices. Enforcing steerable features for all hidden layers offers the advantage of facilitating the construction of the final output space with a designated group representation. For instance, when predicting 3D vectors, the output space needs to correspond to $D^1$ representation. Without adhering to this convention, it would be necessary (and challenging) to track the group representations across all hidden spaces in order to construct such output space.

We provide a formal definition of an equivariant operation as follows:

**Definition 1** *A equivariant operation $f$ takes several steerable features $\{h_1 \in H_1, \cdots, h_k \in H_k\}$ as input, and outputs a steerable feature $h_o = f(h_1, \cdots, h_k)$, s.t.*

$$D^{H_o} h_o = D^{H_o} f(h_1, \cdots, h_k) = f(D^{H_1} h_1, \cdots, D^{H_k} h_k). \tag{4}$$

Some equivariant operations, such as *Linear Combination*, *Layer Normalization*, and *Gate* (Schütt et al., 2017; Weiler et al., 2018), only take a single feature as input. These operations serve as self-interactions and introduce non-linearity in equivariant neural networks. However, in this study, we are primarily interested in operations that take two or more features as input, as they enable modeling of node-to-node or node-edge interactions in a graph. Broadly speaking, there are two types of such operations, that is *tensor product* and *scalarization*.

## 2.3 TENSOR PRODUCT

When an function is mathematical equivariant but does not satisfy definition in 1, the group representation $D$ on the output space is not a direct sum of Wigner-D matrices, and thus cannot be used as equivariant operation. However, since Wigner-D matrices $D^l$ are irreducible representations of $SO(3)$ group, any representation $D$ can be transformed into the direct sum of $\{D^{l_i}\}$ with the change of basis. Therefore, an equivariant operation can take two steps: first apply a mathematical equivariant function to the input features $f_1(h_1, h_2) \rightarrow h_{mid}$; then apply a linear transform to translate it back into steerable features.

The tensor product method serves as an implementation of this concept. When we compute the tensor product of two features $h_{mid} = h_1^{l_1} \otimes h_2^{l_2}$, a new representation $D^{l_1} \otimes D^{l_2}$ is formed over the output space $V^{l_1} \otimes V^{l_2}$. A remarkable property of the Wigner-D matrix is that the tensor product $D^{l_1} \otimes D^{l_2}$ can be efficiently decomposed into a direct sum of Wigner-D matrices $D^{l_i}$ using the Clebsch-Gordan (CG) coefficients. These coefficients, denoted as $C_{(l_1,m_1)(l_2,m_2)}^{(l,m)}$, rely solely on the values of $l_1$, $l_2$, $l_i$, and indices $m_1, m_2$ and is independent of the specific value of $h$. This convenience makes the tensor product a practical method for designing equivariant operations. A tensor product operation $f_1(h_1, h_2) \rightarrow h_o = h_o^{l_1} \oplus h_o^{l_2} \oplus \cdots$ and each $h_o^{l_i}$ can be written as Equation 5, where $w^{l_i, l_1, l_2}$ is a scalar coefficient and can be derived via a MLP. We illustrate the process of tensor product in Figure 1.

$$h_{o,m}^{l_i} = w^{l_i, l_1, l_2} \sum_{m_1=-l_1}^{l_1} \sum_{m_2=-l_2}^{l_2} C_{(l_1,m_1)(l_2,m_2)}^{(l_i,m)} h_{1,m_1}^{l_1} h_{2,m_2}^{l_2} \tag{5}$$

The tensor product method can effectively manage interactions between vectors of different types. However, a significant drawback of this approach is its computationally expensive nature, particularly when dealing with large values of $L$. It has been demonstrated that the computational complexity of the tensor product method scales as $O(L^6)$ (Passaro & Zitnick, 2023).

## 2.4 SCALARIZATION

A special case of an equivariant operation arises when all input and output features are scalars or type-1 vectors. Consider a set of steerable features $h_1, h_2, \ldots, h_k$ that share the same group representation. Since these representations consist of unitary matrices, $(Dh_i)^T(Dh_j) = h_i^T D^T D h_j = h_i^T h_j$ remains invariant when transforming the input coordinates. The scalarization method presents a straightforward algorithm for this scenario. In this equivariant operation, the input vector is first scalarized, followed by MLPs, and the results are aggregated along the original directions of the input vectors. We illustrate the process in Figure 1. Scalarization is an effective method for modeling the interaction between vectors. However, one significant limitation of scalarization is that it cannot change the feature space of input.

## 3 EQUIVARIANT BASIS AND SCALAR INTERACTION

We have presented two equivariant operations, tensor product and scalarization. While the tensor product method provides a comprehensive approach for handling different type-$l$ features, it is burdened by high computational costs. On the contrary, the scalarization method is efficient but cannot model interactions between different type-$l$ features. Therefore, our objective is to devise a more efficient equivariant operation that can interact between different type-$l$ features and surpasses the limitations of the tensor product. To accomplish this, we begin by examining equivariant operations from the perspective of equivariant basis expansion, and see how the tensor product method and scalarization method are derived with equivariant basis in special cases. Then we introduce the idea to reduce the computational cost in equivariant operation by REBIF, and propose our algorithm, scalar interaction.

## 3.1 EQUIVARIANT BASIS

Let us consider an equivariant operation denoted as $f$, which takes steerable features $h_1 \in H_1, \cdots, h_k \in H_k$ as its input. For the simplicity, we use a feature $h_{\text{in}}$ to denote the collection of all the input features and define the group action on all the inputs simultaneously, as show in Equation 4, with $D^{H_{\text{in}}}(g)h_{\text{in}}$. We consider the output $h_o \in V^l$ is a type-$l$ vector. We first define the equivariant basis of the equivariant operation $f$.

**Definition 2** *An equivariant basis for the equivariant operation $f : H_{\text{in}} \rightarrow H^l$ is a collection of equivariant mappings $\{e_1(h_{\text{in}}), e_2(h_{\text{in}}), \ldots, e_d(h_{\text{in}})\}$ that are equivariant with the group:*

$$D^l(g)(e_i(h_{\text{in}})) = e_i(D^{H_{\text{in}}}(g)(h_{\text{in}})), \tag{6}$$

*and every equivariant operation $f : H_{\text{in}} \rightarrow H^l$ can be expanded using this equivariant basis into $d$ scalar value functions:*

$$f(h_{\text{in}}) = \sum_{i=1}^{d} g_i(h_{\text{in}})e_i(h_{\text{in}}), \tag{7}$$

*where $g_i(h_{\text{in}})$ is invariant under input transformations, i.e., $g_i(D^{H_{\text{in}}}(g)(h_{\text{in}})) = g_i(h_{\text{in}})$.*

In this work, we do not discuss the general method of obtaining an equivariant basis. However, we demonstrate that both the tensor product and scalarization methods are based on such an equivariant basis, and they are special cases of the implementation of Equation 7. We first show the relationship between scalarization method and equivariant basis.

**Proposition 1** *Let $f$ be an equivariant operation with respect to the $O(3)$ group, taking scalars $s_1, s_2, \ldots$, and type-1 vectors $u_1, u_2, \ldots, u_k$ as input. Then, one of the equivariant basis is $\{u_i\}$, and Equation 7 can be written as*

$$f(u_1, u_2, \ldots) = \sum_{i=1}^{k} g_i(s_1, s_2, \ldots, u_1, u_2, \ldots)u_i, \tag{8}$$

*where the functions $g_i$ are invariant under the $O(3)$ group.*

Equation 8 leads to the scalarization method, which substitutes the input vectors $u_i$ for the equivariant basis. We adopt the approach proposed by (Villar et al., 2021) for proving Proposition 1, and the detailed proof can be found in Appendix A.1. While this substitution is complete only in the $O(3)$ setting rather than $SO(3)$, the scalarization method remains effective in modeling 3D data and shows promising performance in related tasks.

The motivation behind the tensor product method can also be described using an equivariant basis. The earliest instances of the tensor product method were introduced in the context of steerable convolution on irregular point clouds (Thomas et al., 2018) and regular 3D grids (Weiler et al., 2018). In these scenarios, the equivariant operation involves two inputs: a 3D vector $x$ and a steerable feature $h$. To construct the convolution in Equation 9, the function $f$ must take the form $f(x, h) = \kappa(x)h$ for

$$[\kappa \cdot h](x) = \int \kappa(\boldsymbol{x} - \boldsymbol{x'})h(x)dx = \int f(\boldsymbol{x} - \boldsymbol{x'}, h(x))dx. \tag{9}$$

Under this condition, Proposition 2 holds:

**Proposition 2** *For an equivariant operation $f$ with respect to $SO(3)$ group that takes a type-1 vector $x$ and a steerable feature $h$ as input, and has the form $f(x, h) = \kappa(x)h$, then one of the equivariant basis is $e_i(x, h)$, computed with the tensor product of $h$ and the spherical harmonics $Y^J(x)$. And Equation 7 can be written as*

$$f(h_{\text{in}}) = \sum_{i=1}^{d} g_i(||x||)e_i(x, h). \tag{10}$$

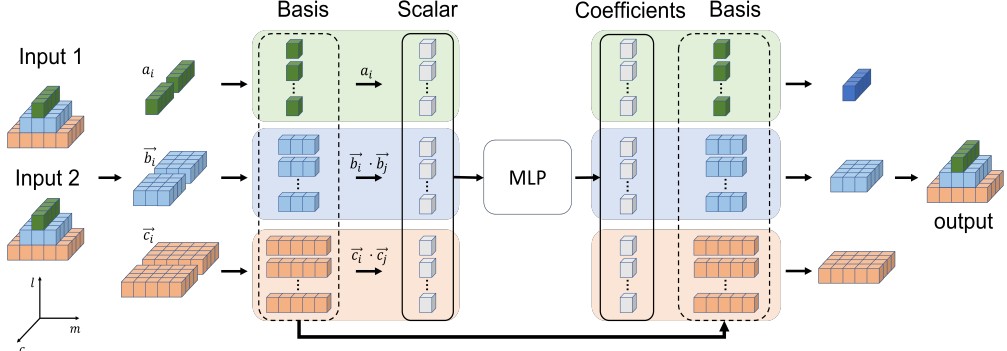

Figure 2: The workflow of Scalar Interaction. It takes two steerable features as input and extracting the fragments as the basis for different subspaces $V^l$. Then, the inner product is computed between fragments of the same dimension, which is utilized to generate the coefficients. The output is computed by taking the linear combination of the basis vectors using these coefficients.

The proof of this proposition is provided in Appendix A.2, which subsequently yields the formula for the tensor product as described in (Weiler et al., 2018).

Similar to the scalarization method, the formulation of tensor product method is complete only within this specific context. There exist equivariant operations that cannot be represented using the tensor product method, such as $f(x, h) = ||x||||h||h$. Nevertheless, the tensor product remains a powerful operation utilized in numerous equivariant networks. Additionally, the introduction of non-linear operations further enhances the expressive capacity of the tensor product method.

## 3.2 REPLACE EQUIVARIANT BASIS WITH INPUT FEATURE

Definition 2 provides a perspective on equivariant operations, by decomposing them into two steps: constructing an equivariant basis and developing invariant scalar functions (coefficients). From this point of view, scalarization method is efficient than tensor product method since it does not compute new basis in the operation. They replace the equivariant basis with components of the input feature. While the current scalarization method is limited to accepting only scalars and type-1 vectors as inputs, we recognize that this approach eliminates the need for computing the equivariant basis and can be readily extended to incorporate steerable features with different type-$l$ vectors. We refer to this method as *Replacing Equivariant Basis with Input Feature* (REBIF for brevity). The formal definition of REBIF is as follows:

**Definition 3** *REBIF. For an equivariant operation $f : H_{\text{in}} \to V^l$, REBIF involves substituting the equivariant basis with all the fragments $h^l \in V^l$ from the collection of input features $h_{\text{in}}$.*

Another key motivation for constructing equivariant neural networks using REBIF arises from the observation of a common pattern in existing equivariant neural networks. Firstly, in most equivariant networks, there is no introduction of new $l$-type features beyond the first layer. Secondly, equivariant neural networks typically have a channel size $c$ that is significantly larger than $2l + 1$. This ensures a high probability that the fragments within the input features can span the entire output space. We provide numerical analysis supporting this claim in subsection 5.2.

## 3.3 SCALAR INTERACTION NETWORK

REBIF serves as a general framework for reducing the computational complexity associated with equivariant operations. Various approaches exist for constructing the invariant function $g_i$ in Equation 7. In this study, we propose a straightforward and intuitive architecture known as Scalar Interaction. This architecture computes scalar values of the input features by taking inner products between vectors of the same type-$l$, and employs a MLP to determine the coefficients $g_i(h_{\text{in}})$. The framework of Scalar Interaction is depicted in Figure 2. Scalar Interaction offers a concise formula and enables message interaction between fragments of different type-$l$ without relying on the tensor

product method. It reduces the computational complexity from $O(L^6)$ to $O(L^3)$. Further details regarding computational complexity and expressiveness are discussed in Appendix B.

## 4 RELATED WORKS

In this section, we focus on equivariant neural networks that work on irregular 3D data.

**Scalarization.** The idea of scalarization is first proposed in SchNet (Schütt et al., 2017) and DimeNet (Gasteiger et al., 2020). SphereNet (Coors et al., 2018), PaiNN (Schütt et al., 2021) follows these work. EGNN (Satorras et al., 2021) propose a flexible paradigm with is E(n) equivariant. GMN (Huang et al., 2022) extends it with multi vectors setting.

**Tensor Product.** Tensorfield Networks (TFN), (Thomas et al., 2018) and NequIP (Batzner et al., 2022) use graph neural network with equivariant linear message passing. SEGNN (Brandstetter et al., 2022b) introduces non-linear message passing with steerable MLP. SE(3)-Transformer (Fuchs et al., 2020) uses dot product to construct invariant attention for message passing, followed by Torch-MD (Thölke & Fabritiis, 2022) and EQGAT (Le et al., 2022). Equiformer (Liao & Smidt, 2022) propose MLP attention an non-linear message for construct more expressive transformer.

**Regular Representation.** Another line of work construct equivariant network by *lifting* and group convolution (Finzi et al., 2020; Hutchinson et al., 2021). These method also face a trade-off between computational complexity and performance as a result of discretization and sampling.

## 5 EXPERIMENTS

We implement scalar interaction based on the e3nn library. We implement two models. The first model uses scalar interaction layer to compute message between the neighbor nodes and do equivariant message passing between neighbor nodes. This model is called SiNet. Since scalar interaction cannot create new $l-$ type features if the input doesn't contain such features, we adopt a single tensor product layer to construct the node embedding. The tensor product operation here is performed between scalars and other features, thus its computational complexity is the same as a Linear layer, which will not limit the speed of SiNet. Furthermore, we notice that the state-of-the-art results on QM9 prediction task are achieved by models with more complex structure like attention. Therefore, we propose an equivariant transformer based on scalar interaction for this task. We name this model *SiFormer*. The implementation details is in Appendix D.

### 5.1 THE RUNNING SPEED

**The speed of scalar interaction operation.** As one of the most significant benefits of SINet, the computation complexity is much smaller that tensor product method, especially when the input feature has a large maximum $l$. Therefore, we made some experiments to compare the running speed of SINet with tensor product based method. We test a single operation with input $h_1 \in n_c V_0 \oplus n_c V_1 \oplus \cdots \oplus n_c V_L$ and $h_2 \in V_0 \oplus V_1 \oplus \cdots \oplus V_L$ since this is a normal setting in equivariant neural networks, where $h_1$ is the node feature and thus have multiple channels, and $h_2$ denotes the edge features and are usually a single channel. The results are show in Figure 3. Furthermore, we construct a model using SINet to show its performance, the result is included in Appendix D.1.

### 5.2 THE COMPLETENESS OF BASIS

One of the motivations behind REBIF is the expectation that the input features will span a "large" subspace of each vector space $V^l$, facilitated by the substantial channel size relative to the vector dimension. To evaluate the completeness of REBIF, we record the inputs of the Scalar Interaction at various layers during the QM9 experiments (see subsection 5.3) and analyze the extent to which the input features can serve as a basis. To this end, we construct $L$ matrices, each containing all the $l$-type features $h_i^l$, organized as a matrix $H^l = (h_1^l, \cdots, h_{c_l}^l) \in \mathbb{R}^{(2l+1) \times c_l}$. Subsequently, we compute the metric: $r(l) = \frac{\text{rank}(H^l)}{2l+1}$, where $\text{rank}(\cdot)$ determines the rank of a matrix, and $2l+1$

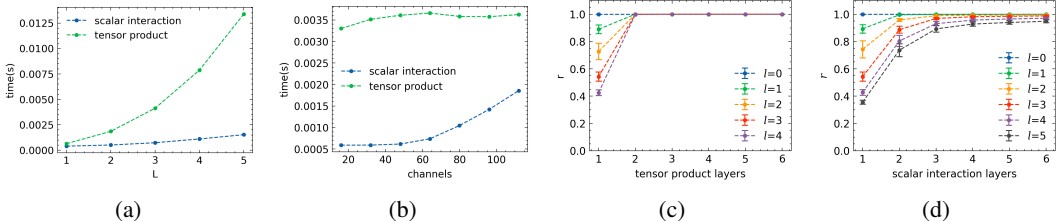

Figure 3: The comparison between the tensor product and scalar interaction methods is presented in terms of running speed and the completeness of the basis. In (a), we examine the running time of both methods for different values of $L$, while keeping the number of channels $n_c$ fixed at 64. (b) depicts the running time for varying $n_c$ with a maximum $L$ of 3. Furthermore, we analyze the completeness metric $r$ of node features for different tensor product layers in (c), and for Scalar Interaction layers in (d).

denotes the dimension of $h_i^l$. $r(l)$ quantifies the ratio of linearly independent input features to the total number of basis vectors. If the input features can span the entire vector space, $r$ will equal 1. We calculate this metric for all nodes, obtaining the mean and variance. The results are presented in Figure 3. Importantly, we observe that in tensor product-based methods, features spanning a substantial portion of the vector space are achieved after the first layer. Since tensor product operations can generate new basis vectors in each layer, all types of features rapidly span the entire space within two layers. In SINet, new basis vectors are not generated through equivariant operations starting from the second layer. However, due to message passing, which aggregates information from a node's neighbors, the completeness continues to increase as the number of layers grows. More discussion is provided in Appendix D.2.

## 5.3 QM9: REGRESSION ON SCALAR LABELS

**Dataset.** The QM9 dataset consists of quantum chemical properties for small molecules composed of up to 29 atoms with atomic types including H, C, N, O, and F. Each datum includes 3D coordinates and atom types. This dataset is commonly used to evaluate the ability of equivariant networks to regress the chemical properties for each molecule. We adopt the data splitting method of () and use 110k, 10k, and 11k molecules for the training, validation, and testing sets, respectively. We train our model by minimizing the mean absolute error (MAE) between predictions and labels. Details of the training are provided in the Appendix D.3.

Table 1: Result of regression on QM9 dataset.

| Methods | Task
Units | $\alpha$
$a_0^3$ | $\Delta\varepsilon$
meV | $\varepsilon_{\text{HOMO}}$
meV | $\varepsilon_{\text{LUMO}}$
meV | $\mu$
D |
|---|---|---|---|---|---|---|
| NMP (Gilmer et al., 2017) [†] | | .092 | 69 | 43 | 38 | .030 |
| SchNet (Schütt et al., 2017) | | .235 | 63 | 41 | 34 | .033 |
| Cormorant (Anderson et al., 2019) [†] | | .085 | 61 | 34 | 38 | .038 |
| LieConv (Finzi et al., 2020)[†] | | .084 | 49 | 30 | 25 | .032 |
| DimeNet++ (Klicpera et al., 2020) | | **.044** | 33 | 25 | 20 | .030 |
| TFN (Thomas et al., 2018)[†] | | .223 | 58 | 40 | 38 | .064 |
| SE(3)-Transformer (Fuchs et al., 2020) [†] | | .142 | 53 | 35 | 33 | .051 |
| EGNN (Satorras et al., 2021) [†] | | .071 | 48 | 29 | 25 | .029 |
| PaiNN (Schütt et al., 2021) | | **.045** | 46 | 28 | 20 | .012 |
| TorchMD-NET (Thölke & Fabritiis, 2022) | | .059 | 36 | 20 | 18 | **.011** |
| SphereNet (Coors et al., 2018) | | .046 | **32** | 23 | 18 | .026 |
| SEGNN (Brandstetter et al., 2022b) [†] | | .060 | 42 | 24 | 21 | .023 |
| EQGAT (Le et al., 2022) | | .053 | 32 | 20 | 16 | **.011** |
| Equiformer (Liao & Smidt, 2022) | | .046 | **30** | **15** | **14** | **.011** |
| SINet (Ours) | | .058 | 41 | 22 | 20 | .023 |
| SiFormer (Ours) | | .056 | 33 | **18** | **15** | .014 |

[†] denotes using different data partition

**Results.** We evaluated SINet's performance on the first five targets, and the results are presented in Table 1. Our experiments show that SINet outperforms TFN and EGNN, indicating that scalar interaction is a more powerful method for equivariant operations. The ablation study in Table 3a shows the benefits of introducing higher order type of vectors. SiFormer, which uses scalar interaction to construct a graph transformer, achieves results comparable to other state-of-the-art methods, providing evidence that SINet can serve as a building block for more complex architectures.

## 5.4 N-BODY SYSTEM: REGRESSION ON 3D VECTOR LABELS

Table 2: Result of regression on N-body system.

| Methods | SE(3)-Tr. | TFN | NMP | Radial Field | EGNN | SEGNN | SINet |
|---------|-----------|-------|-------|--------------|-------|-----------|--------|
| MSE | .0244 | .0155 | .0107 | .0104 | .0070 | **_.0043_** | **.0044** |

**Dataset.** N-body system dataset consists of the dynamical simulation of 5 charged particles in three dimension space. The regression target is the particle positions after 1,000 timesteps, which is $1-$type vectors. We build the experimental setting follow the work of (Brandstetter et al., 2022a). We use the relative position $x_i$, velocity $v_i$ and the norm of velocity as input, embed edge features as spherical harmonics $Y_l^m(\mathbf{x_j} - \mathbf{x_i})$, and construct node attributes with the edge features and the embeddings of velocity. The details are provided in Appendix D.4.

**Results.** The results of SINet and other baseline models are shown in Table 2. SINet demonstrates comparable results to SEGNN while offering the advantage of being approximately four times faster, primarily due to its avoidance of tensor product operations. In addition, we conduct an ablation study to investigate the impact of using different values of $L$ in SINet, which is presented in Table 3b. The results indicate that the best performance is attained when $L = 1$. This observation aligns with the findings from the SEGNN experiment, suggesting that it may be attributed to the fact that the targets primarily consist of $l = 1$ vectors.

Table 3: Ablation studies of SINet on the max type $L$

(a) QM9

| $L$ | 1 | 2 | 3 | 4 |
|------|----|----|----|----|
| MAE (meV) | 28 | 26 | **23** | **23** |

(b) N-body system

| $L$ | 1 | 2 | 3 | 4 | 5 |
|------|---------|-------|-------|-------|--------|
| MSE | **.0044** | .0047 | .0049 | .0049 | **.0046** |

## 6 LIMITATIONS AND DISCUSSION

The proposed Scalar Interaction method utilizes REBIF and avoids generating new vector bases. Consequently, if the fragments of the input features fail to span the output space, the expressiveness of the method is limited. One possible approach to alleviate this limitation is to introduce more tensor product operations at the first layer of the neural network. Furthermore, since the scalar interaction method calculates the inner product of input feature fragments, the information retained in these scalars determines the universality of the formula. The first fundamental theorem for $O(d)$ (Villar et al., 2021) demonstrates that inner products contain all the necessary information for constructing $O(d)$ invariant functions for type-1 vectors. However, for other type-$l$ vectors, there may be a loss of information. Future research can explore alternative methods for constructing scalars.

## 7 CONCLUSION

In this study, we introduce Scalar Interaction as a novel equivariant operation that utilizes REBIF to reduce the computational complexity associated with tensor product operations. We employ Scalar Interaction as a building block to construct equivariant models, namely SINet and SiFormer. We evaluate the performance of these models on a real-world dataset, demonstrating the effectiveness of scalar interaction.

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
