# OpenReview forum: "E(3) Equivariant Scalar Interaction Network"
_ICLR.cc/2024/Conference — Submitted to ICLR 2024_

### Official Review · Reviewer_UiX2 · 2023-10-29

**Soundness:** 2 fair
**Presentation:** 3 good
**Contribution:** 3 good
**Rating:** 3
**Confidence:** 3

**Summary:**

This paper proposes a new equivariant operation called Scalar Interaction for building efficient equivariant neural networks. The key ideas are:
- Provide a unified view of equivariant operations as equivariant basis decomposition.
- Introduce Replace Equivariant Basis with Input Features (REBIF) to avoid computing new bases.
- Propose Scalar Interaction to compute interactions between input feature fragments via inner products.
- Build a Scalar Interaction Network (SINet) using these concepts that can handle higher-order features with lower complexity than tensor products.

The method is evaluated on quantum chemistry and n-body physics tasks, demonstrating comparable performance to prior equivariant networks, with increased efficiency.

**Strengths:**

- The equivariant basis decomposition perspective provides theoretical unification of different techniques.
- REBIF is a simple but effective idea.
- Scalar Interaction enables handling higher-order features with low complexity.

**Weaknesses:**

-  The proposed method appears to be a hybrid of tensor product and scalarization methods, as it first uses the tensor product to generate high-order input features and then processes these features through scalar interaction blocks (resembling scalarization).

- The motivation for this paper could benefit from further justification, given that the assumption that tensor product methods are more accurate than scalarization methods is not clearly supported. For example, Table 1 shows that the scalarization method EGNN outperforms several tensor product methods.

- The Scalar Interaction approach directly uses fragments of input features as the equivariant basis, rather than computing new bases like tensor product methods do. Although this approach is faster, it might compromise expressiveness and universality for certain tasks compared to methods that explicitly compute new bases like tensor products.

**Questions:**

- More analysis of the expressiveness limitations of scalarization models, to support the motivation.
- Are there any ablation studies to examine the effectiveness of the proposed modifications?

---

### Official Review · Reviewer_6zpN · 2023-10-30

**Soundness:** 2 fair
**Presentation:** 3 good
**Contribution:** 3 good
**Rating:** 6
**Confidence:** 4

**Summary:**

The authors present a unifying formulation for equivariant neural networks that combine aspects of tensor product layers and scalarization. Both the tensor product and scalarization equivariance can be written as constructing an equivariant basis and using invariant scalar functions to find the coefficients for the basis vectors. In scalarization, unlike the tensor product layers, the basis is immediately available as the input features themselves are used as part of the basis. However, this method does not apply to higher-order representations beyond type-1 vectors.

The proposed Scalar Interaction Network (SINet) uses a similar idea -- Replacing Equivariant Basis with Input Feature (REBIF) -- to construct the equivariant basis, but generalizes it to higher order representations. The architecture involves computing scalar values using inner products between input representations of the same type and then passing all the scalars from all the types to an MLP to compute the invariant coefficients. Thus, interaction between the types is achieved and equivariance is maintained without costly operations. Experiments on QM9 and N-body dynamics show that the proposed idea is competitive with state-of-the-art, better than EGNN and TFN, and computationally much faster than using tensor product layers, especially when the maximum order of the representation is high.

**Strengths:**

1. The authors present their ideas well in a simple intuitive way, and the paper is well-written.

2. The ideas seem intuitive and simple to implement.

3. The architecture is more general compared to the scalarization idea that applies only to type-1 vectors.

4. The architecture, while being simple, still yields very good performance on standard geometric graph datasets.

**Weaknesses:**

1. In Table 3(b), L=1 has the best performance. This seems to suggest that just type-1 vector interaction used in scalarization is sufficient for the N-body dynamics, thus demonstrating no added benefit of the SINet idea. Am I understanding this correctly?

2. While Table 1 shows improved results compared to EGNN and TFN, how SINet can do better than TFN is unclear. How are the architectures being compared? Is it based on having the same computational budget?

These experimental details are the main weaknesses in the paper, in my opinion.

**Questions:**

No additional questions.

---

### Official Review · Reviewer_ZnMo · 2023-11-04

**Soundness:** 2 fair
**Presentation:** 1 poor
**Contribution:** 2 fair
**Rating:** 3
**Confidence:** 4

**Summary:**

In this work, the authors presented SINet, a hybrid approach combining both tensor product operations and scalarizations to build efficient equivariant operations for geometric data. The authors first present an equivariant basis decomposition framework to unify equivariant operations including tensor products and scalarizations. By using this framework, the authors proposed Scalar Interaction to construct the mapping between different type-L features with quadratic complexity. Experiments are conducted to demonstrate the efficiency and empirical performance of the proposed SINet.

**Strengths:**

1. The efficient issue of tensor product based equivariant operations is of great interest to the broad community of geometric deep learning.
2. The quadratic complexity of Scalar Interaction between type-L features is appealing given the $\mathcal{O}(L^6)$ complexity of standard CG tensor product.

**Weaknesses:**

1. **Regarding the capacity of the proposed approach**: although the scalar interaction approach is faster than the standard CG tensor product, its expressive power and universal approximation capability are limited due to the input features basis.

2. **Regarding the acceleration**: From Figure 3, the acceleration ratio decays with more channels are used. Besides, although the complexity is reduced from $\mathcal{O}(L^6)$ to $\mathcal{O}(L^2)$, the empirical acceleration ratio does not seem to have orders of magnitude faster.

3. **Regarding the empirical performance**: (1). From Table 3, SINet with steerable features in larger L does not consistently yield better performance. In this sense, the significance of scalar interactions is doubtful because the brought acceleration does not enable achieving better performance (even worse, then why not to use the scalarization approaches?); (2). Besides, as in EGNN/SEGNN paper, they provide both MSE and time evaluation on N-body system. The readers would be also curious about the efficiency comparison between scalarization and Scalar Interaction because the latter is a hybrid approach involving scalarization. (3) The performance of Table 1 is not appealing compared to previous baselines. (4) It would be better to conduct ablation studies on the proposed approach.

**Questions:**

1. In Appendix A.1, could you provide an example on the group action $\tilde{Q}$?
2. Could you provide efficiency comparison between Scalar Interaction and Scalarization approaches?
3. Could you provide ablation studies on the proposed SINet/SIFormer?

---

### Meta-Review · Area_Chair_D7Yj · 2023-12-03

**Metareview:**

This work presents a conceptual framework that unifies the equivariant operations via equivariant basis decomposition. Based on this framework, the authors then generalize the idea of replacing the equivariant basis with input features to design efficient equivariant operations capable of modeling different type-features. The reviewers raised multiply concerns about this work in the review process, including the capacity of the model and the empirical advantages. The authors didn't provide any rebuttal. Given those unaddressed concerns, I recommend rejection.

**Justification For Why Not Higher Score:**

See the metareview.

**Justification For Why Not Lower Score:**

N/A

---

### Decision · Program_Chairs · 2024-01-16

Reject